# Spheroids Composed of Reaggregated Neonatal Porcine Islets and Human Endothelial Cells Accelerate Development of Normoglycemia in Diabetic Mice

**DOI:** 10.3390/cells14050366

**Published:** 2025-03-02

**Authors:** Mohsen Honarpisheh, Yutian Lei, Antonia Follenzi, Alessia Cucci, Cristina Olgasi, Ekaterine Berishvili, Fanny Lebreton, Kevin Bellofatto, Lorenzo Piemonti, Antonio Citro, Francesco Campo, Cataldo Pignatelli, Olivier Thaunat, Elisabeth Kemter, Martin Kraetzl, Eckhard Wolf, Jochen Seissler, Lelia Wolf-van Buerck

**Affiliations:** 1Medizinische Klinik und Poliklinik IV, Diabetes Zentrum, Klinikum der Universität München, LMU Munich, 80336 Munich, Germany; mohsen.honarpisheh@med.uni-muenchen.de (M.H.); leiyutian1234@gmail.com (Y.L.); jochen.seissler@med.uni-muenchen.de (J.S.); 2Department of Health Sciences, University of Piemonte Orientale, 28100 Novara, Italy; antonia.follenzi@med.uniupo.it (A.F.); alessia.cucci@uniupo.it (A.C.); 3Department of Translational Medicine, University of Piemonte Orientale, 28100 Novara, Italy; cristina.olgasi@med.uniupo.it; 4Tissue Engineering and Organ Regeneration Lab, University of Geneva, Department of Surgery, CH-1211 Geneva, Switzerland; ekaterine.berishvili@unige.ch (E.B.); fanny.lebreton@unige.ch (F.L.); kevin.bellofatto@unige.ch (K.B.); 5Diabetes Research Institute, IRCCS Ospedale San Raffaele, 20132 Milan, Italy; piemonti.lorenzo@hsr.it (L.P.); citro.antonio@hsr.it (A.C.); cataldo.pignatelli@hsr.it (C.P.); 6Department of Endocrinology, Università Vita-Salute San Raffaele, 20132 Milan, Italy; 7Centre International de Recherche en Infectiologie, Université Claude Bernard Lyon I, 69364 Lyon, France; olivier.thaunat@chu-lyon.fr; 8Department of Nephrology Transplantation and Clinical Immunology, Edouard Herriot Hospital, Hospices Civils de Lyon, 69003 Lyon, France; 9Molecular Animal Breeding and Biotechnology, Gene Centre and Department of Veterinary Sciences, Ludwig-Maximilians-Universität München, 80539 Munich, Germany; kemter@lmb.uni-muenchen.de (E.K.); ewolf@lmb.uni-muenchen.de (E.W.); 10German Center for Diabetes Research (DZD), 85764 Neuherberg, Germany

**Keywords:** xenotransplantation, neonatal porcine islet-like cell clusters, reaggregated cell clusters, blood outgrowth endothelial cells

## Abstract

The engraftment of transplanted islets depends on the rapid establishment of a novel vascular network. The present study evaluated the effects of cord blood-derived blood outgrowth endothelial cells (BOECs) on the viability of neonatal porcine islets (NPIs) and the post-transplant outcome of grafted NPIs. Dispersed NPIs and human BOECs were reaggregated on microwell cell culture plates and tested for their anti-apoptotic and pro-angiogenic capacity by qRT-PCR and immunohistochemistry. The in vivo functionality was analyzed after transplantation into diabetic NOD-SCID IL2rγ^−/−^ (NSG) mice. The spheroids, which contained reaggregated neonatal porcine islet cells (REPIs) and BOECs, exhibited enhanced viability and a significantly elevated gene expression of VEGFA, angiopoetin-1, heme oxygenase-1, and TNFAIP3 (A20) in vitro. The development of normoglycemia was significantly faster in animals transplanted with spheroids in comparison to the only REPI group (median 51.5 days versus 60 days) (*p* < 0.05). Furthermore, intragraft vascular density was substantially increased (*p* < 0.01). The co-transplantation of prevascularized REPI-BOEC spheroids resulted in superior angiogenesis and accelerated in vivo function. These findings may provide a novel tool to enhance the efficacy of porcine islet xenotransplantation.

## 1. Introduction

Islet cell transplantation represents an established, potentially life-saving therapy for type 1 diabetes [1]. However, its clinical application is limited by the shortage of organ donors and the rapid loss of islet mass in the first days after transplantation, which results in failure to achieve insulin independence with a single islet preparation in most patients [2]. Therefore, patients with type 1 diabetes desperately require new therapeutic solutions that can better replace beta cell function.

It has been demonstrated that small islets or size-controlled pseudoislets reaggregated from dispersed human and murine islet cells (IC) exhibit enhanced functionality in vitro and in vivo [3,4,5,6]. This is due to the fact that in large islets, the centrally located cells undergo ischemic death prior to re-establishment of vascularization from the recipient’s vessels [7]. In accordance with this concept, additional improvement of viability and functionality was evidenced through the utilization of mixed hybrid cell clusters comprising ICs and endothelial cells, which have the capacity to expedite the revascularization of islets. Blood outgrowth endothelial cells (BOECs) are cells derived from the circulating endothelial progenitors found in the peripheral blood, umbilical cord blood (CB), or bone marrow. They promote neovascularization by incorporation into damaged and injured vessels and by the release of pro-angiogenic factors [8,9]. Previous studies have demonstrated that the coating of murine or human islets with BOECs prior to transplantation enhanced the revascularization and function of human [10], mouse [11,12], and rat [13] islet grafts. Furthermore, enhanced beta cell function, reduced glycemia, and increased beta cell proliferation were observed following marginal mass transplantation of hybrid murine IC-BOEC spheroids [14,15]. These effects were attributed to the release of anti-inflammatory and proangiogenic factors, cell–cell interaction, and the production of matrix proteins that mimic islet cell microenvironment [10,13,16,17].

To date, only a limited number of studies have examined the potential beneficial effects of BOECs on porcine islets. Kang and colleagues have demonstrated that the coating of adult pig islets with BOECs results in enhanced vascularization, which in turn leads to improved glycemic control following transplantation into diabetic nude mice [17]. Neonatal porcine islets (NPIs) are the most promising candidate cells for xenotransplantation, as they are easier to isolate than adult islets and demonstrate greater resistance against inflammation and hypoxia [18,19]. A disadvantage of using islet cells from neonatal pigs is that they are immature and require several weeks for maturation after transplantation [6]. Given the importance of neighboring cells in modulating cellular behaviors, the present study analyzed the potential improvement of the NPI cell product by generating 3D spheroids composed of islet cells and BOECs. The findings of this study demonstrate that interactions between islet cells and BOECs enhance islet viability and increase vascularization of transplanted NPIs. Engineering islet-BOEC spheroids may represent an effective strategy to improve the efficiency of porcine islet xenotransplantation.

## 2. Materials and Methods

### 2.1. Animals

Donor pancreata were explanted from German Landrace hybrid piglets. Immunodeficient NOD-SCID *IL2rγ^−/−^* (NSG) mice were obtained from The Jackson Laboratory (strain 005557) and kept under standard barrier (SPF) conditions. All the animal procedures were performed according to the German Animal Welfare Act and Directive 2010/63/EU after approval of the Animal Ethics Committee.

### 2.2. Isolation and Validation of Cord Blood-Derived Blood Outgrowth Endothelial Cells (BOECs)

Human umbilical cord blood samples were collected from healthy newborns. Mononuclear cells were isolated by Ficoll density gradient centrifugation. The cells were cultured at 37 °C under 5% CO_2_ in a humidified incubator in Endothelial Cell Growth EGM-2 medium (Lonza, Basel, Switzerland) on 25 cm^2^ flasks coated with 50 μg/mL VitroCol^®^ (Advanced Biomatrix, Carlsbad, CA, USA). After 48 h of culture, non-adherent cells and debris were aspirated. Adherent cells were cultured in EGM-2 medium, which was changed every other day. BOECs from passages 5–8 were analyzed for the expression of endothelial markers (hCD31, hCD144, CD146) and the absence of leukocyte markers (CD45, CD14) by flow cytometry. For further characterization, BOECs were seeded on 12-well plates coated with Matrigel (Corning, NY, USA) (2.5 × 10^5^ cells/well) according to the manufacturer’s instructions. Tube formation was documented under a microscope after 24 h and 48 h.

### 2.3. Islet Isolation and Generation of Spheroids Consisted of REPI-BOEC Cells

We isolated NPIs from full-term piglets at the age of 2–5 days as previously described [20]. After collagenase digestion, cells were cultured in basal islet culture (B-IC) medium (RPMI 1640 (PAN-Biotech, Aidenbach, Germany), 2% human serum albumin (Takeda, Konstanz, Germany), 10 mM nicotinamide, 20 ng/mL exendin-4 (Merck, Darmstadt, Germany), and 1% antibiotic–antimycotic (Thermo Fisher Scientific, Germering, Germany). Single-cell preparation of NPIs was performed on day 4 through gentle dissociation with TrypLE^TM^ Select solution (Thermo Fisher Scientific) as recently described [6].

BOECs from passages 5–8 were harvested by incubation in TrypLE^TM^ Select solution, washed with RPMI 1640 medium, and mixed with dissociated porcine islet cells (ICs) (ratio 75% ICs/25% BOECs). The cells were then seeded on 24-well Sphericalplate 5D^®^ (Kugelmeiers, Erlenbach, Switzerland) (concentration of 750 islet cells plus 250 BOECs per microwell in 80% B-IC medium/20% EGM-2 medium) for 48 h to form spheroids. For control, cell clusters consisting only of ICs were cultured under the same conditions. After 2 days of co-culture, spheroids were harvested and used for TUNEL assay and transplantation.

### 2.4. Cluster Composition and Cell Viability

On day 2 after cluster formation, REPIs and REPI-BOECs were collected from the plates and fixed in 4% paraformaldehyde (PFA) for 2 h. Then, clusters were embedded in Epredia™ HistoGel™ Specimen Processing Gel (Thermo Fisher Scientific) and re-fixed in 4% PFA overnight. After a washing step, samples were embedded in paraffin, and 2 µm sections were subjected to immunostaining for hCD31. Cell death was assessed by TUNEL assay (Thermo Fisher Scientific) according to the manufacturer’s protocol.

### 2.5. Static Glucose-Stimulated Insulin Secretion

REPIs and REPI-BOEC spheroids were harvested after 48 h of incubation in Sphericalplate 5D plates. The clusters were washed three times in PBS and twice in Krebs–Ringer buffer (KRB: 135 mM NaCl, 4.8 mM KCl, 1.2 M Mg_2_SO_4_, 1.2 mM KH_2_PO_4_, 1.3 mM CaCl_2_, 5 mM NaHCO_3_, 10 mM HEPES, and 0.5% BSA pH 7.4). Subsequently, 100 clusters were incubated in duplicate in 24-well plates in KRB buffer containing low glucose (2.8 mM) for 1 h and then challenged with low glucose or high glucose (20.0 mM) for 1 h at 37 °C, 5% CO_2_ [20]. The supernatants were collected and stored at -80 °C. Insulin concentration in the supernatant was measured by porcine insulin ELISA (Mercordia, Uppsala, Sweden) according to the manufacturer’s instructions. The stimulation index (SI) was calculated by the following formula: insulin concentration^high glucose^/insulin concentration^low glucose^.

### 2.6. Quantitative Reverse Transcription Polymerase Chain Reaction

Total RNA extraction from cell samples and cDNA synthesis were performed as recently described [20]. Relative mRNA levels of human vascular endothelial cell growth factor (*VEGFA*), human angiopoetin-1 (*ANG1*), porcine heme oxygenase (HO)-1 (*HMOX1*) (a major ROS-scavenger enzyme in pancreatic beta cells with antioxidant, antiapoptotic, and anti-inflammatory effects [21,22]), and antiapoptotic protein A20 (*TNFAIP3*), which function as a cellular defense mechanism against inflammation and oxidative stress [23,24], were quantified on a MaxPro-Max3000P Real-time PCR system (Stratagene, La Jolla, CA, USA) using SsoFast™ EvaGreen^®^ (Bio-Rad, Laboratories, Feldkirchen, Germany) and the following reaction conditions: 10 min, 95 °C; 40 cycles at 95 °C for 10 s, 60 °C for 20 s, followed by a melting curve stage [20]. Relative gene expression was normalized to the expression level of porcine beta-2-microglobulin (*B2M*) or human glyceraldehyde 3-phosphate dehydrogenase (*GAPDH*) using the comparative 2^−ΔCt^ method. The final normalized quantity was calculated using the formula 2^−ΔΔCt^. All primer sequences are listed in Appendix A.

### 2.7. Transplantation

Diabetes was induced in NSG mice by intraperitoneal injection of 180 mg/kg streptozotocin (Merck, Darmstadt, Germany). Diabetic NSG mice (blood glucose levels > 350 mg/dL) received equal numbers of 1500 IEQ REPIs or 1500 IEQs REPI-BOEC spheroids under the left kidney capsule. To compare development of normoglycemia with standard NPI transplantation, a third group of animals was transplanted with 3000 IEQ unmodified NPIs as described recently [6]. Diabetic mice that exhibit blood glucose levels >300 mg/dL were treated with insulin glargine. Restoration of normoglycemia, defined as sustained random non-fasting blood glucose levels < 120 mg/dL over a 16-week observation period, represented the primary endpoint. Glucose tolerance was assessed 10–14 days after the development of normoglycemia by intraperitoneal glucose tolerance test (IPGTT) using 2 g glucose/kg body weight. Blood samples were taken from the tail vein at 0 and 10 min for analysis of porcine insulin by ELISA (Mercodia, Uppsala, Sweden). To demonstrate that normoglycemia was mediated by the transplanted tissue, the graft-bearing kidneys were removed in three transplanted animals per group. The reoccurrence of diabetes (blood glucose level was >350 mg/dL) after uninephrectomy indicated that normoglycemia was graft dependent.

### 2.8. Immunohistochemistry

Paraffin sections of spheroids and the graft bearing kidneys were stained for all transplanted cells (mouse anti-human pan-cytokeratin, 1:100), beta cells (guinea pig anti-insulin, 1:400) (Agilent-Dako Waldbronn, Germany), endothelial cells (rabbit anti-mouse CD31, 1:100, Cell Signaling Technology, Danvers, MA, USA), mouse anti-human CD31 1:20 (Agilent-Dako), transcription factors essential for beta cell specificity (rabbit anti-homeodomain transcription factor Nk6 homeobox 1 (Nkx6.1, 1:2000, Cell Signaling Technology), rabbit anti-v-Maf musculoaponeurotic fibrosarcoma oncogene homologue A (MAFA, 1:100, Bethyl Laboratories, Montgomery, AL, USA) and proliferating cells (rabbit anti-Ki-67, 1:10,000, Proteintech, Planegg-Martinsried, Germany). As secondary antibodies, HRP-conjugated or alkaline phosphatase-conjugated anti-guinea pig IgG, anti-mouse IgG, anti-rabbit IgG, and a polymer-based detection system (Immpress HRP Polymer Kit, Vector Laboratories, Burlingame, CA, USA) were used. Fuchsin plus substrate chromogen (Agilent-Dako) or 3,3′-diaminobenzidine (DAB) were used as chromogens (Merck). For double staining against human and mouse CD31 FITC anti-mouse IgG (1:20, Dako-Agilent) and Alexa594 anti-rabbit IgG (1:500, Invitrogen, Carlsbad, CA, USA) antibodies were used. Quantification of mouse CD31^+^, human CD31^+^, Ki-67^+^, pan-cytokeratin^+^, MAFA^+^, and Nkx6.1^+^ cells in the grafted area was performed by Qupath software (version 0.3.2) using slide scans generated by the uScope MXII slide scanner (Microscope International, Dallas, TX, USA).

### 2.9. Flow Cytometry

Islets and spheroids were dispersed into single cells by digestion with TrypLE^TM^ Select solution. Cells were washed with PBS + 10% fetal calf serum (FCS) and incubated with Fc-Block (anti-mouse CD16/CD32) for 10 min followed by staining with fluorochrome-labeled antibodies against hCD31-APC, hCD144-PE, hCD146-FITC, hCD14-FITC, and hCD45-APC (BD Biosciences, Heidelberg, Germany). Antibodies were incubated for 30 min at 4 °C, washed twice with PBS + 10% FCS, and analyzed on a BD LSRFortessa flow cytometer (BD Biosciences) with FlowJo software version 10.4 (TreeStar, Ashland, OR, USA).

### 2.10. Statistical Analysis

Data are expressed as mean and standard deviation (SD). Statistical analyses were performed using the two-tailed Student’s *t*-test, the one-way ANOVA (AUC analysis), or log-rank test (diabetes reversal) by Prism software version 9.2 (GraphPad, San Diego, CA, USA). The AUC was calculated by Prism software using trapezoidal rules. *p*-values < 0.05 were considered significant.

## 3. Results

### 3.1. Formation and Characterization of REPI-BOEC Spheroids Before and After Transplantation

BOECs, isolated from human cord blood, expressed characteristic endothelial cell (EC) markers such as hCD31, hCD144, and hCD146, but not the leukocyte markers hCD14 or hCD45. BOEC function was tested by the formation of capillary tube-like structures in a tube formation assay (Appendix A). These BOECs were used to generate 3D clusters with dispersed NPIs on Sphericalplate 5D microwell plates. The morphology of the spheroids composed of NPIs and BOECs showed the formation of round-shaped clusters (Figure 1A). Immunohistochemical staining demonstrated that BOECs covered the surface of the spheroids, and many cells were well integrated into the core of the clusters (Figure 1A). While prevascularization does not affect in vitro glucose-dependent insulin secretion (GSIS) (Figure 1B), the TUNEL assay revealed that cell death was significantly reduced in spheroids as compared to the REPI-only group, resulting in about 8% more viable cells at the day of transplantation (Figure 1C). We then analyzed the expression of genes known to be involved in the stress response of beta cells. The relative gene expression of porcine *HMOX1* (3.1 ± 0.6-fold at day 1 and 1.9 ± 0.7-fold at day 3, *p* < 0.001) and porcine *TNFAIP3* (*A20*) (1.7 ± 0.2-fold at day 1, 2.8 ± 1.2-fold at day 3, *p* < 0.01) was significantly increased in spheroids as compared to REPIs alone (Figure 1D).

### 3.2. BOECs Accelerate the Achievement of Full Graft Function

To analyze the impact of BOECs on the in vivo maturation and function of REPIs, we transplanted equal IEQs of REPIs and REPI-BOEC spheroids into diabetic NGS mice. Notably, the diabetes reversal rate was increased (100% vs. 87.5%), and the time to develop euglycemia (median 51.5 vs. 60.0 days) was significantly reduced in the group transplanted with spheroids (*p* < 0.02) (Figure 2A). The area under the curve (AUC) for glucose (0–120 min during IPGTT) and glucose-dependent insulin secretion (0 min, 10 min) were similar in both groups (Figure 2B–D). These data indicate that transplantation of REPIs with incorporated BOECs did not negatively impact porcine islet function but resulted in a significantly better post-transplant outcome in terms of time needed to control glycemia. Upon removal of the graft-bearing kidney, all uninephrectomized mice developed hyperglycemia.

Notably, quantification of Ki-67 positive cells (proliferation marker) within the grafts at day 7 after transplantation showed no significant differences between the REPI and the REPI-BOEC groups, suggesting that the beneficial effect of BOECs is not related to islet cell proliferation (Figure 2E). The number of grafted cells immunostained for nuclear Nkx6.1 or nuclear MAFA was similar in both groups at posttransplant day 7 (Figure 2E).

### 3.3. BOECs Increase Intragraft Vascular Density After Transplantation

Based on the results of the transplantation experiments, we examined in vivo vascularization after spheroid transplantation next. Vessel density within the graft was significantly increased in spheroid-transplanted mice compared to the REPI-only group (Figure 3A,B). The majority of CD31-positive cells within the graft area were of murine origin, suggesting that de novo formed vessels were mainly of recipient origin. However, double staining for mouse and human CD31 revealed that some human ECs in the graft, derived from the incorporated BOECs, are integrated into functional blood vessels in direct contact with mouse ECs (Figure 3A, right panel). Finally, qRT-PCR was performed to evaluate the expression of proangiogenic factors. Using species-specific primers, the expression of human *VEGFA* and *ANG1* was detected in the REPI-BOEC spheroids, indicating that BOECs express key mediators to induce angiogenesis (Figure 3C).

## 4. Discussion

Proper islet vascularization is critical for graft survival after islet transplantation [25,26]. This study demonstrated for the first time that NPI-BOEC spheroid formation improves islet cell viability and glycemic control after transplantation into diabetic NSG mice as compared to mice transplanted with islet cells alone.

Islets are highly vascularized mini-organs whose function depends not only on well-organized endocrine cells but also on specific interactions of endocrine cells with endothelial and mesenchymal cells, which play important roles in regulating islet function [27,28]. Numerous studies have reported that BOECs have beneficial effects on angiogenesis and promote neovascularization in models of vascular hypoxia such as myocardial infarction or limb ischemia [29,30] and in islet transplantation at different transplantation sites. Better glycemic control was observed after co-transplantation of human BOECs with human islets [10], or transplantation of a mixture of mouse islets with mouse BOECs [12], rat islets with human BOECs [13,16], and adult porcine islets with human BOECs [17]. In the present study, we report an improved outcome in terms of time to develop normoglycemia after transplantation of spheroids composed of porcine neonatal islets and human BOECs, suggesting increased cell survival after transplantation. Our histological quantification of proliferation markers and transcription factor signature of mature beta cells argues against the hypothesis that BOECs induce higher islet cell proliferation or directly increase beta cell maturation in the early pretransplant period (day 7). Further studies with longitudinal analysis of beta cell maturation and proliferation are needed to assess BOEC effects at later time points after transplantation.

BOECs may exert various effects on islet cells mediated by direct paracrine factors that improve beta cell survival and promote rapid revascularization. Our in vitro studies show a beneficial interaction between NPIs and BOECs. Islet isolation and cultivation as well as hypoxic conditions after transplantation can induce endoplasmic reticulum (ER) stress and ROS formation that reduces beta cell viability [31,32]. We observed an increased viability of islet cells after spheroid generation, which may be partly explained by an improved resistance to oxidative stress [33]. Indeed, here we detected an upregulation of *HMOX1* and *TNFAIP3* gene expression in islets after spheroid formation, which both play a protective role by dampening inflammation and inhibiting islet apoptosis. Our findings are in agreement with previous studies reporting an increased survival and improved outcome after transplantation of islets expressing A20 or HO-1 [21,24].

In addition, we showed that rather than directly forming the neovessels in transplanted islets, BOECs expressed *VEGFA* and *ANG1*, which are potent proangiogenetic factors that activate multiple signaling cascades involved in the attraction and proliferation of recipient EC progenitors [34,35]. Previous studies have demonstrated that expression of *VEGFA* or *ANG1* in transplanted islets increases vascularization and islet blood flow, leading to improved islet survival and better glucose control in recipient animals [28]. VEGFA can induce new vessel formation in or around transplanted islets [27,36,37]. Consistent with this finding, we observed a significant increase in vessel density after transplantation in the spheroid group. Immunohistological studies at 7 days and 4 months suggested that incorporated BOECs may not contribute primarily to direct graft neovascularization but rather promote the recipient neoangiogenesis through the secretion of proangiogenic factors, recruitment of recipient endothelial cells, and differentiation into functional vessels. Previous studies have also reported increased beta cell proliferation in islet grafts after co-transplantation of BOECs [17], a finding that we did not observe in the present study. This discrepancy may be explained by the islet cell source, species differences, and different characteristics of the isolated BOECs.

In conclusion, the incorporation of BOECs into REPIs prior to transplantation represents a novel strategy to protect neonatal porcine islet cells, resulting in better engraftment and improved islet survival and function. These data suggest that REPI-BOEC spheroid formation could become a clinically applicable strategy to improve porcine islet xenotransplantation.

## Figures and Tables

**Figure 1 cells-14-00366-f001:**
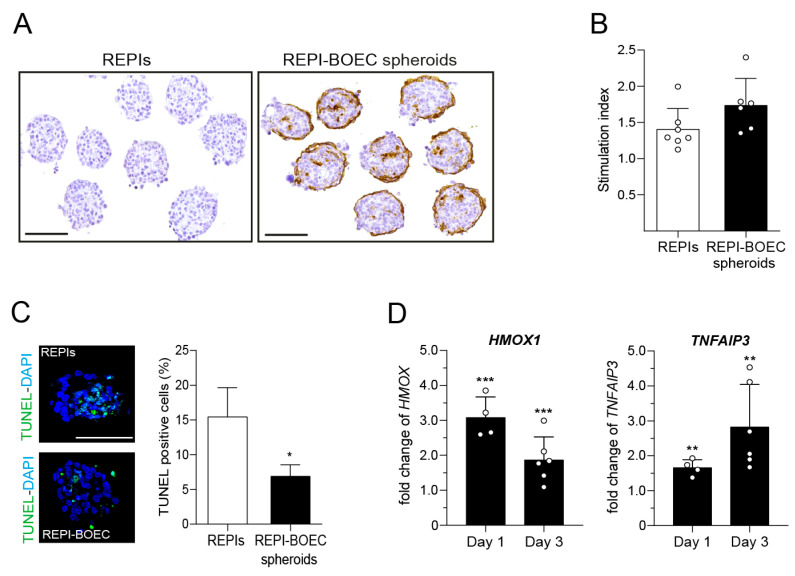
Generation and phenotypic characterization of REPIs and REPI-BOEC spheroids. (**A**) Phase contrast images of composite spheroids and REPIs generated in Sphericalplate 5D 24-well plates and stained for hCD31 (brown). Scale bar = 100 µm. (**B**) In vitro beta cell function measured by static glucose-stimulated insulin secretion (n = 6–7). (**C**) Cell viability at day 2 after cluster formation as determined by TUNEL assay (n = 3). (**D**) Real-time PCR analysis of porcine heme oxygenase-1 (*HMOX-1*) and A20 (*TNFAIP3* gene expression at day 1 and 3 after spheroid formation (n = 4–6). Data shown are mean ± SD and represent the results of at least three independent experiments. * *p* < 0.05, ** *p* < 0.01, *** *p* < 0.001 REPI-BOEC spheroids vs. REPIs. Scale bar = 100 µm.

**Figure 2 cells-14-00366-f002:**
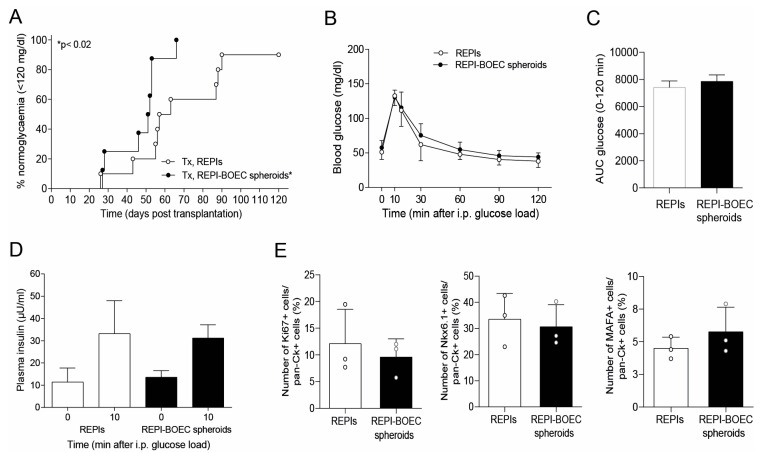
Transplantation with REPI-BOEC spheroids improved diabetes reversal. (**A**) Life table analysis of diabetes reversal in diabetic NSG mice transplanted with 1500 IEQ REPIs (open cycles n = 10) or 1500 IEQ REPI-BOEC spheroids (black cycles, n = 8). Spheroid transplantation reduced the time to develop normoglycemia (*p* < 0.02 according to the log-rank test). (**B**–**D**) Intraperitoneal glucose tolerance test (IPGTT) performed 8–14 days after the development of persistent normoglycemia was similar in both transplant groups as assessed by measurement of blood glucose profiles (**B**), glucose clearance (AUC glucose 0–120 min) (**C**), and insulin secretion at 0 and 10 min after glucose challenge (**D**). (**E**) Quantification of grafted cells (day 7 after transplantation) immunostaining for Ki67 (cell proliferation), Nkx6.1 (marker of endocrine progenitor cells and mature beta cells), or MAFA (mature beta cells) and pan-cytokeratin (staining of all transplanted porcine cells) revealed no difference in groups transplanted with REPIs alone or with REPI-BOEC (n = 3).

**Figure 3 cells-14-00366-f003:**
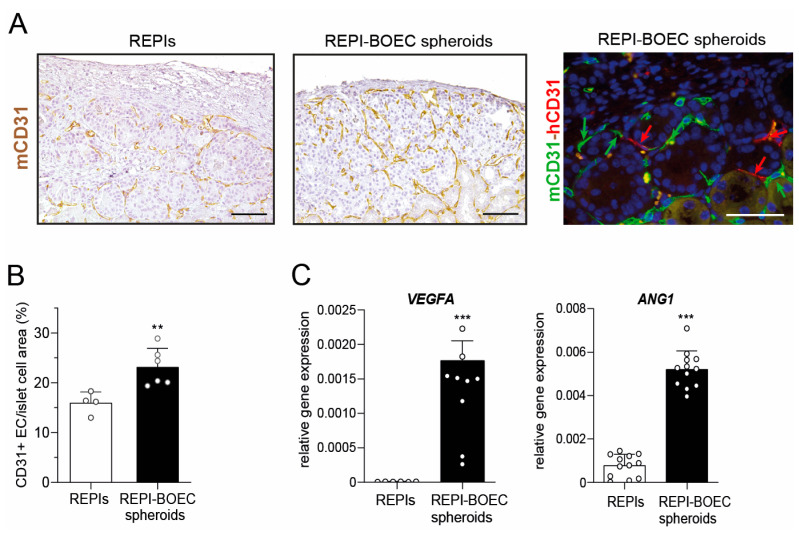
Spheroids display proangiogenic capacity in vitro and improve revascularization of transplanted grafts in vivo. (**A**) Detection of blood vessels in the grafted area at the end of the observation period (posttransplant week 16). Representative images show a higher intragraft vascular network in mice transplanted with REPI-BOEC spheroids compared to REPIs alone. Left: Immunostaining for mouse CD31 (brown). Right: Double immunostaining for mouse CD31 (green) and human CD31 (red) showing the integration of human ECs into the vascular network. Scale bar = 100 µm. (**B**) Quantification of the vessel density (number of mCD31 positive cells) within the graft area. (**C**) Gene expression of the human angiogenic factors *VEGFA* and *ANG1* in REPIs and REPI-BOEC spheroids relative to human *GAPDH* detected by qRT-PCR. Data are presented as mean ± SD and are generated from at least three independent experiments. ** *p* < 0.01, *** *p* < 0.001.

## Data Availability

The data presented in this study are available in the article and Appendix A.

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
