# Peer review of "Spheroids Composed of Reaggregated Neonatal Porcine Islets and Human Endothelial Cells Accelerate Development of Normoglycemia in Diabetic Mice"

_cells, 2025, doi:10.3390/cells14050366_

Round 1
Reviewer 1 Report
Comments and Suggestions for Authors
The authors attempted to elucidate the usefulness of neonatal porcine islet spheroids including human endothelial cells in xenotransplantation model. In comparison with porcine islet spheroids, they showed longer engraftment with hyper vascularization. The human endothelial cells promoted the vascularization derived from mainly mouse. Angiogeneic factors including VEGFA and angiopoietin 1 might contribute to the vascularization.
I have some comments and questions.
1. Could you show the data or evidence why human endothelial cells included in porcine islet spheroids avoided apoptosis of porcine islet cells. Are there any rational reasons and data? (Figure 1C)
2. All the qPCR data should be changed to 2-deltadelta CT data. (Figure 1D and 3C)
3. I have no idea why the xenotransplantation of porcine islet spheroids including human endothelial cells was obtained longer engraftment in comparison with porcine islet spheroids without human endothelial cells, despite of no improvement of insulin secretion, no improvement in glucose tolerance test, and no cellular proliferation. Please discuss why. In my opinion, porcine islet spheroids including human endothelial cells might be tough against hypoxia and apoptosis, and therefore more islet spheroids were succeeded to be engrafted. If so, the number and volume of engrafted islets spheroids were higher than spheroids without endothelial cells. Please show the proof in revision. Furthermore, histological assessment for apoptosis of islet grafts in early timing (till 2 days after xenotransplantation) are recommended. The measured bloods glucose data is also important. (Figure 2)
4. I think assessment of porcine islets (not reconstructed spheroids, just isolated from donor pancreas) is important. The process of constructing spheroids might impair isolated porcine islets. Therefore, there might be functional benefits (releasing insulin, for example) in isolated islets comparing with islet spheroids without endothelial cells. How about including porcine islet (not spheroids) xenotransplantation in this study? I have some interest in comparison between porcine islet spheroids including human endothelial cells and isolated porcine islets. Please discuss further examinations. (Figure 2)
Reviewer 2 Report
Comments and Suggestions for Authors
The manuscript requires a more comprehensive analysis of BOEC-mediated NPI maturation and function. However, there are several key areas that require attention to strengthen the study's conclusions.
The first concern is the lack of detailed blood glucose profiles for individual mice post-transplantation. This data is crucial for a comprehensive assessment of the treatment's efficacy and consistency. Without it, it's difficult to fully evaluate the impact of the proposed method on glycemic control over time.
While the authors claim that BOECs enhance NPI maturation in vivo, they do not sufficiently elaborate on the underlying mechanisms or molecular pathways involved. To strengthen the scientific argument, a more thorough explanation of how BOECs facilitate islet maturation and integration into the host environment is necessary.
Furthermore, the current presentation of beta cell proliferation data is limited and potentially misleading. The study only provides data from a single early timepoint (7 days post-transplantation), which does not accurately reflect the long-term maturation and functionality of NPIs, which typically require several weeks to engraft and function effectively. To address this, the authors should include data from later timepoints, such as 2 and 4 weeks post-transplantation.
Building on this, the study would significantly benefit from a comprehensive timeline of beta cell proliferation and function extending beyond the first two weeks. This broader perspective would better illustrate the long-term effects of BOECs on islet maturation and glycemic control. By providing this extended timeline, the authors can offer a clearer picture of the transplantation method's efficacy in promoting sustained beta cell function and glucose homeostasis. This more complete dataset would allow for a more accurate evaluation of the potential of BOECs in enhancing islet xenotransplantation outcomes and lend greater credibility to the study's conclusions.
Comments on the Quality of English LanguageThe overall quality of English in the manuscript is good, with clear and coherent writing throughout most sections. However, there are some minor issues that, if addressed, would further improve the clarity and readability of the text.
In the abstract, there is an unnecessary article "an" in the phrase "an improved in vivo maturation" that should be removed. The introduction contains a complex sentence about ischemic death in islets that could be simplified for better comprehension.
The Materials and Methods section would benefit from small adjustments, such as adding "markers" after "endothelial" when discussing protein expression, and changing "by digestion" to "through digestion" when describing cell dispersion processes.
In the Results section, a few sentences could be rephrased for improved clarity. For instance, "versus only REPI" could be changed to "compared to REPI alone", and "by log-rank test" to "according to the log-rank test".
The Discussion section generally reads well, but could be further refined by changing "and to enhance" to "while enhancing" in one sentence, and simplifying another by replacing "may become" with "could become".
Lastly, there is an inconsistency in the use of "REPIs" versus "REPI" in figure captions that should be addressed for consistency.
These suggestions are minor and do not detract significantly from the overall quality of the manuscript. Implementing these changes would further refine the text and enhance its clarity and professionalism. The authors have demonstrated a good command of scientific English, and these small improvements will help to polish the final product.
Reviewer 3 Report
Comments and Suggestions for Authors
In their study, Honarpisheh et al. describe how re-aggregation of neonatal porcine islets with blood outgrowth endothelial cells improves time to restoration of normoglycemia in a transplantation model, likely due to increased islet cell viability and maturation following improved vascularization. Overall, the manuscript is of good quality and I agree with its conclusions. I only have a few suggestions.
1) Because protein stability is a major determinant of HIF1α abundance and hence signalling, can the authors detect differences HIF1α protein levels via immunoblot? This could be done under HIF1α-inducing conditions such as high glucose (using suitable positive controls) as a diabetes-relevant stimulus. Alternatively, are HIF1α target genes lower expressed in REPI-BOEC spheroids?
2) In their discussion (line 320/321), enhanced maturation is suggested as a mechanism for improved time to normoglycemia. Can the authors stain their histological REPI/REPI-BOECs samples (7 d and 16 weeks post-transplant) for a few beta-cell maturation markers to support their interpretation?
3) Line 255-256: “Upon removal of the graft-bearing kidney, all uninephrectomized mice developed hyperglycemia.” Could this data be included, either as supplement or in Fig. 2?
Minor:
Fig. 3 A, right panel: There appear to be what looks like co-staining of mouse and human CD31, can the authors comment on this apparent contradiction?
For clarity, please mention the time point at which IPGTTs (Fig. 3 B-D) and analysis of proangiogenic factors (Fig. 3) were performed (16 weeks post-transplant?).
Fig. 1 (legend): D=C, E=D.
Line 272: “spheroids“
Round 2
Reviewer 1 Report
Comments and Suggestions for Authors
The revised manuscript is improved and I think it is in sufficient level for publication to Cells.
Reviewer 3 Report
Comments and Suggestions for Authors
I like to thank the authors for taking the time to perform additional experiments to support their conclusions. All points raised previously have been adequately addressed.
Comments on the Quality of English LanguageJust minor typos detected, e.g. line 183 "factot", line 333f "and and", Fig. S2C: "transplanttaion".